# Psoriasis and Antimicrobial Peptides

**DOI:** 10.3390/ijms21186791

**Published:** 2020-09-16

**Authors:** Toshiya Takahashi, Kenshi Yamasaki

**Affiliations:** Department of Dermatology, Tohoku University Graduate School of Medicine, Sendai, Miyagi 980-8574, Japan; kyamasaki@med.tohoku.ac.jp

**Keywords:** psoriasis, antimicrobial peptides, β-defensin, S100 proteins, cathelicidin (LL-37), DAMPs, NETs, plasmacytoid dendritic cells, Th17

## Abstract

Psoriasis is a systemic inflammatory disease caused by crosstalk between various cells such as T cells, neutrophils, dendritic cells, and keratinocytes. Antimicrobial peptides (AMPs) such as β-defensin, S100, and cathelicidin are secreted from these cells and activate the innate immune system through various mechanisms to induce inflammation, thus participating in the pathogenesis of psoriasis. In particular, these antimicrobial peptides enhance the binding of damage-associated molecular patterns such as self-DNA and self-RNA to their receptors and promote the secretion of interferon from activated plasmacytoid dendritic cells and keratinocytes to promote inflammation in psoriasis. Neutrophil extracellular traps (NETs), complexes of self-DNA and proteins including LL-37 released from neutrophils in psoriatic skin, induce Th17. Activated myeloid dendritic cells secrete a mass of inflammatory cytokines such as IL-12 and IL-23 in psoriasis, which is indispensable for the proliferation and survival of T cells that produce IL-17. AMPs enhance the production of some of Th17 and Th1 cytokines and modulate receptors and cellular signaling in psoriasis. Inflammation induced by DAMPs, including self-DNA and RNA released due to microinjuries or scratches, and the enhanced recognition of DAMPs by AMPs, may be involved in the mechanism underlying the Köbner phenomenon in psoriasis.

## 1. Introduction

Psoriasis is a representative autoimmune or inflammatory skin disorder characterized by well-delineated, raised areas of red or salmon-pink papulosquamous plaques covered by white or silvery scales [1]. Psoriasis shows a diverse prevalence across worldwide populations: 1.5–3% in Europeans [1], 0.05–3% in Africans, and 0.1–0.5% in Asians [2]. Several triggering factors have been linked with an exacerbation of psoriasis, such as infection, wounds, obesity, stress, and genetic factors [3], and exposure to certain drugs can induce or exacerbate psoriasis. Strong associations have been documented for beta-blockers, lithium, antimalarial drugs such as chloroquine, interferons, imiquimod, and terbinafine [4], and new associations have been reported for monoclonal antibodies such as tumor necrosis factor (TNF)-α antagonists and anti-programmed cell death protein 1 immune checkpoint inhibitors [4].

Psoriasis affects not only the skin but also nails and joints. Up to 30% of psoriasis patients develop psoriatic arthritis (PsA) [5]. In addition, patients with psoriasis have increased prevalence of cardiovascular risk and metabolic syndrome, such as hyperlipidemia, insulin resistance, diabetes, and obesity [6]. Patients with more severe psoriasis have greater odds of metabolic syndrome than those with milder psoriasis [7]. Epidemiologic associations between psoriasis and gastrointestinal diseases, kidney diseases, malignancy, infection, and mood disorders have been proven. Shared inflammatory pathways, cellular mediators, genetic susceptibility, and common risk factors are hypothesized to be contributing factors [8]. Psoriasis is thus a disease caused by systemic inflammation which presents unique skin symptoms of inflammatory keratosis.

In cellular and molecular pathology, psoriasis is regarded as a T cell-mediated skin disease, involving both T-helper (Th)1 and Th17 cells to promote inflammation by producing cytokines, including TNF-α interferon (IFN)-γ, interleukin (IL)-12, IL-17A, IL-22, and IL-23 [9,10]. Antibodies against TNF-α (etanercept, adalimumab, and infliximab), IL-12 (ustekinumab), IL-23 (guselkumab, risankizumab, and tildrakizumab), IL-17A (secukinumab and ixekizumab), and IL-17AR (brodalumab) have shown clinical efficacy in improving skin and joint conditions [6] and are now indispensable for the treatment of psoriasis. Psoriasis pathogenesis does not completely depend on T cell-mediated adaptive immune disorder, and innate immune cells, neutrophils, and plasmacytoid dendritic cells (pDCs) are also involved in pathogenesis. The disease often reoccurs shortly after withdrawal of these DC or T cell-targeted monotherapies, suggesting that blocking adaptive immune activation alone is insufficient to treat psoriasis [11,12,13]. In addition to studies on adaptive and innate immune cells and inflammatory cells, investigations on psoriatic epidermal keratinocytes have shown that abnormally differentiated and proliferated keratinocytes are sources of abundant inflammatory cytokines and chemokines in skin lesions. Furthermore, recent studies on the molecular pathogenesis of psoriasis revealed that keratinocytes produce antimicrobial peptides and proteins (AMPs) that activate immune cells via multiple mechanisms. This article summarizes how AMPs participate in psoriasis pathogenesis and discusses the implications of AMPs as alarmins in psoriasis treatment.

## 2. AMPs Expressed in Skin and Dermatoses

AMPs have essential roles in skin immunity, enabling epithelial surfaces to cope with many microbial challenges [14]. AMPs are evolutionarily ancient innate immune effectors and are synthesized by almost all plants and animals [15]. More than 1800 AMPs have been identified and more than 20 are found in human skin [16]. In general, AMPs are small peptides composed of 12–50 amino acid residues and have positive charges and amphipathic structures [16]. These features allow AMPs to interact with negatively charged phospholipid head groups and the hydrophobic fatty acid chains of microbial membranes, killing select microorganisms by disrupting the microbial membrane and releasing cytosol components [17,18]. AMPs show anti-microbial activity against a diverse range of skin pathogens, including Gram-negative and -positive bacteria, fungi, viruses, and parasites [16].

Psoriatic lesions highly express AMPs such as cathelicidin, β-defensins, S100 proteins, RNase 7, lysozyme, elafin, and neutrophil gelatinase-associated lipocalin [19,20]. Contrary to the term “antimicrobial”, AMPs are not only natural antibiotics that directly kill or inhibit the growth of microorganisms [21] but also modify host inflammatory responses by a variety of mechanisms. AMPs serve in host inflammatory reactions as chemotactic agents, angiogenic factors, and regulators of cell proliferation [16]. Common human skin disorders such as rosacea [22,23,24], atopic dermatitis [25], and psoriasis [26] have been linked to an excessive expression of AMPs. Clearly, these skin diseases cannot be attributed only to microorganisms, and AMPs are involved in the pathogenesis of these dermatoses via host inflammatory reactions partly independently of microorganisms.

### 2.1. Cathelicidin Antimicrobial Peptides (CAMPs)

Cathelicidin was the first AMP identified in mammalian skin [27]. A single cathelicidin antimicrobial peptide gene (CAMP) encodes the precursor protein hCAP18 in humans [28]. hCAP18 is variously cleaved by proteases to generate several active AMPs, including the 37-amino-acid peptide LL-37 [29]. LL-37 is a 37-residue peptide generated from cathelicidin with two leucines at the N-terminus. LL-37 is expressed by various types of cells, such as epidermal keratinocytes, intestine cells, respiratory epithelial cells, neutrophils, T cells, natural killer cells, monocytes, and mast cells [16,30,31]. LL-37 is detectable in skin, trachea, esophagus, intestine, stomach, liver, spleen, and bone marrow, as well as in sweat, saliva, wound fluid, and seminal plasma [32]. In normal skin, keratinocytes produce various AMPs at low levels to defend the skin barrier [33], whereas cathelicidin precursor protein and mature peptide are most abundantly expressed by resident mast cells [34]. Mast cells are typically present around blood vessels and store large amounts of cathelicidin in preformed granules. This localization places AMPs derived from mast cells in an ideal position to resist infections after skin injury and inoculation with pathogens [35]. Kulkarni et al. reported that noncoding RNA can increase adhesion molecules on endothelial cells in the presence of LL-37 [36]. Once inflamed, skin produces cathelicidin through increased expression of CAP18 by keratinocytes and adipocytes [37] and increased local deposition by recruited neutrophils [38,39,40].

As natural antibiotics, AMPs target the essential cell wall or cell membrane structures of microorganisms through enzymatic or nonenzymatic disruption. Simultaneously, AMPs can also function as potent immune regulators by signaling through chemokine receptors and inhibiting or enhancing Toll-like receptor (TLR) signaling [14]. Furthermore, CAMP is triggered both by stimulation from pathogen-associated molecular patterns and by damage-associated molecular patterns (DAMPs), including urea [41] and nucleic acids [42], implying that CAMPs are both anti-microbially and immunologically active.

The expression, secretion, and activity of most AMPs are tightly controlled. Cathelicidins are synthesized as propeptides and are activated into various CAMPs including NL-8, LR-10, KR-10, IK-14, LL-17, LL-23, KR-20, KS-27, KS-30, and LL-37 by serine proteases [43]. However, it is unclear whether those peptides exist in psoriatic skin, except for LL-37. In neutrophils, the propeptides are removed by proteinase [17,44], whereas processing is carried out by kallikreins (KLKs, also known as stratum corneum tryptic enzyme) in keratinocytes [43]. Interestingly, the antimicrobial activities of cathelicidin peptides depend on their size [45]. Processing mechanisms generate the active forms of AMPs in skin. The specificity of each activation mechanism helps prevent potential harmful effects of these proteins on mammalian cell membranes [46]. LL-37 can be a substrate for two irreversible post-translational modifications, citrullination and carbamylation, linked to neutrophil activity [47]. The role of cathelicidins in psoriasis is summarized in Figure 1.

### 2.2. Defensins

Defensins are a type of cationic microbial peptide and contain six conserved cysteine residues that form three pairs of intramolecular disulfide bonds [16,19]. In contrast to the presence of the single human cathelicidin gene CAMP, humans have multiple defensin genes that form several gene clusters. For example, six human α-defensins (human neutrophil peptide [HNP]1–6) have been identified. α-Defensins are mainly produced by neutrophils and Paneth cells. In skin, HNP1, HNP2, and HNP3 have been identified from lesional psoriatic scale extracts [20]. Approximately 90 β-defensin genes have been identified in mice and humans. Four human β-defensins (hBD-1–4) have broad spectrum antimicrobial activity and immune-modulating functions and are expressed in epithelia and peripheral blood cells. HBD-1 is constitutively expressed in epithelia but hBD-2–4 are only induced by stimulation with pro-inflammatory cytokines and microbial products [16]. Similar to cathelicidin, β-defensins are expressed as propeptides, although the processing mechanism remains unknown [48]. hBD is induced by TNF-α and IFN-γ, which are highly expressed in psoriasis lesional skin [49]. Interestingly, TNF-α and IL-17A synergistically promote hBD-2 secretion [50] via the induction of transcription factors such as OCT-1, NF-κB, and AP-1 [51]. Although the effects of hBD on psoriasis are largely unknown, Rohrl et al. showed that hBD-2 acts as a ligand for chemokine receptor 6 (CCR6) [52]. The CCR6 signal is known to induce Th17 in psoriatic skin [53], suggesting the induction of Th17 by hBD-2. Furthermore, hBD-2 was identified as a biomarker of IL-17A-driven pathology by comparing protein expression in patients with psoriasis versus that in healthy subjects [54]. Sweeney et al. also showed that mouse β-defensin 14, an ortholog of hBD-3, stimulates Langerhans cells to produce IL-23, resulting in mild psoriasis-like inflammation [55]. Although the involvement of Langerhans cells in psoriasis is controversial, hBD3 may have a similar effect in humans. The role of hBD in psoriasis is summarized in Figure 2.

### 2.3. S100 Proteins

S100 proteins are another group of AMPs important in psoriasis and comprise a family of low molecular weight (9–13 kDa) proteins characterized by the presence of two calcium-binding helix-loop-helix motif [56]. S100 proteins are involved in regulating protein phosphorylation, transcription factors, intracellular Ca^2+^ signaling, cytoskeletal membrane interaction, enzyme activities, cell cycle progression, differentiation, and inflammatory responses. Twenty-one S100 proteins are known, of which S100A7 (psoriasin), S100A8 (calgranulin A), S100A9 (calgranulin B), S100A12 (calgranulin C), and S100A15 have antimicrobial effects and their expression levels are increased in the lesional skin and serum of psoriasis patients [19]. In particular, S100A7 is well studied and was first isolated in psoriatic epidermis [57]. S100A7 is induced by calcium, vitamin D, retinoic acid, bacterial products, TNF-α, IL-17A and IL-22, and is involved in the pathogenesis of psoriasis via its chemotactic activity for neutrophils and CD4^+^ T lymphocytes [58]. Hegyi et al. reported that topical application of calcipotriol decreased the secretion of S100A7 and S100A15, indicating one mechanism by which vitamin D derivatives affect the skin in psoriatic lesions [59]. The role of S100 protein in psoriasis is summarized in Figure 2.

### 2.4. Other AMPs

Shao et al. reported that lipocalin-2 (Lcn2), an AMP derived from keratinocytes and neutrophils, was highly expressed in the lesional skin of psoriatic patients. In vitro, Lcn2 stimulated human neutrophils to produce proinflammatory mediators such as IL-6, IL-8, TNF-α, and IL-1α via a specific receptor, 24p3R, on neutrophils [60]. Elgharib et al. showed statistically significant correlation between serum elafin levels and Psoriasis Area and Severity Index (PASI) score [61].

### 2.5. AMPs from Skin-Commensal Staphylococci Serve as a Skin Barrier to Control Microbiota

AMPs provide defense and resistance to infection by killing pathogenic bacteria. AMPs also determine the microbiota composition and limit access of the microbiota to host tissues. Surprisingly, an important component of the surface antimicrobial shield of the skin is produced by the resident microorganisms themselves. Gram-positive bacteria such as *Lactococcus*, *Streptococcus*, and *Streptomyces* produce factors, known as bacteriocins, which are another type of AMP and inhibit the growth of other bacterial strains and species that could compete for nutrients and other resources. *Staphylococcus epidermidis*, the dominant bacterium cultured from skin microflora, produces the AMP phenol-soluble modulin (PSM)-γ. PSMγ causes membrane leakage in target bacteria, indicating that it functions in a manner similar to that of host-derived AMPs [62]. Interestingly, PSMs are functional in vivo; nanomolar concentrations decreased the survival of group A streptococcus on normal human skin but did not affect the survival of *S. epidermidis* from which the peptide is derived. In addition, PSMs enhance the bactericidal activity of human neutrophils by inducing their neutrophil extracellular traps (NETs) [62], suggesting that human innate immune systems cooperate with commensal bacteria to balance the microbiome via these AMPs.

Another important example of the protective action of *S. epidermidis* in vivo was observed on the surface of the nasal cavity. Nasal colonization by *S. aureus* was inhibited in individuals whose nasal passages were colonized with specific strains of *S. epidermidis* that produce a serine protease capable of destroying biofilms formed by *S. aureus* [63]. For example, a thiolactone-containing peptide produced by *S. epidermidis* blocks the *S. aureus* quorum-sensing system that controls the production of various virulence factors [64]. The selective activity of AMPs produced by commensal organisms may therefore be an important part of the normal host defense strategy against pathogen colonization, with microbe-derived AMPs probably working together with host-derived proteins to establish the composition of the skin surface microbiome.

## 3. Cell-Specific Regulation of AMPs in Psoriasis

In psoriasis skin lesions, keratinocytes, neutrophils, dendritic cells, and T cells have their own roles in generating unique skin symptoms by expressing AMPs, with cell-specific AMP production modulating intracellular and intercellular reactions in psoriasis.

### 3.1. Keratinocytes and AMPs

Various external stimuli result in rapid innate immune responses by keratinocytes, leading to the production of an array of pro-inflammatory cytokines or chemokines such as IFN-β, IL-1β, IL-36, TNF, IL-6, IL-8, IL-25, and CXCL10 [65,66,67]. These cytokines prime and amplify epidermal innate immune signals with the dermal adaptive immune system, contributing to autoimmune activation and psoriasis pathogenesis [42,65,66,68]. Zhang et al. showed that the induction of IFN expression in keratinocytes is one of the earliest innate immune events during skin injury [69]. Keratinocyte-derived IFN-β promotes dendritic cell maturation and subsequent T cell proliferation, leading to psoriatic inflammation [69]. Kabashima’s group reported that conditional deletion of TNF receptor-associated factor 6 (TRAF6) in keratinocytes abrogated dendritic cell (DC) activation, IL-23 production, IL-17 production by γδ T cells, and subsequent IL-17-mediated psoriatic inflammation in an imiquimod psoriasis mice model; furthermore, epidermal TRAF6 was required for the full development of IL-17-mediated inflammation [70]. These studies suggest that the innate immune responses of keratinocytes are essential to prime the autoimmune cascade and drive psoriasis pathogenesis, and type 1 IFN may function as an early initiating factor linking skin wounds with adaptive immune activation that drives psoriasis. Along with the increase of these inflammatory cytokines in psoriasis, keratinocytes in psoriasis patients show overproduction of antimicrobial peptides. Cathelicidin and its human active form, LL-37, likely regulate psoriasis since cathelicidin expression is increased [25,38] and skin infection is decreased in areas affected by psoriasis [25]. LL-37 further stimulates keratinocytes to produce IL-36 and other cytokines as alarmins [71]. Thus, the production of cathelicidin forms the cytokine feedback loop in psoriasis.

Harder et al. isolated human β-defensin (hBD)-2 [49] and hBD-3 [72] from lesional skin scales on psoriasis patients and from cultured keratinocytes. Hollox et al. showed a significant association between higher genomic copy number for hBD genes and risk of psoriasis [73]. Qiao et al. reported that the production of hBD and another AMP, LCN2, increased significantly in keratinocytes following mechanical stretch [74]. In addition, Bhatt et al. showed that sustained secretion of S100A7 from differentiated keratinocytes is dependent on the downregulation of caspase-8, and that IL-1α is necessary and sufficient to induce S100A7 secretion [75]. Collectively, AMPs from keratinocytes are essential for the pathogenesis of psoriasis.

### 3.2. Neutrophils, Neutrophil Extracellular Traps (NETs) and AMPs

Neutrophil extracellular traps (NETs) are of central importance in psoriasis. NETs are a complex of self-DNA and proteins, including LL-37 and proteases released from neutrophils after NET-specific cell death (NETosis). NETs were originally reported to be triggered by bacterial components and inflammatory mediators such as IL-8 and type I IFN. NETs are “webs” that capture bacteria and thus exhibit antibacterial action [76,77]. Lambert et al. showed that NETosis occurs in psoriatic skin lesions to induce Th17, and that mutation of TRAF3IP2, a psoriasis risk gene mutation, enhanced Th17 induction [78]. This raises the possibility that neutrophils are involved in the pathogenesis of psoriasis via NETosis following Th17 activation. On the other hand, Herster et al. reported that LL-37 in complex with RNA induced the release of NETs via TLR8/TLR13-mediated cytokine, whereas LL-37 in complex with DNA did not [79]. Moreover, AMPs have been detected in mast cell extracellular traps (MCETs), which are similar to NETs [80].

### 3.3. Dendritic Cells and AMPs

Plasmacytoid dendritic cells (pDCs) are a unique subpopulation which infiltrate psoriatic skin rapidly [81,82]. The high expression of intracellular receptors, including Toll-like receptors (TLRs), allows pDCs to sense viral or autologous nucleic acid released from damaged cells. The pDCs then produce a large quantity of IFN-α, which initiates the autoimmune cascade [81,83]. The activation of pDCs precedes myeloid/conventional DC (mDC) or T cell activation [81], suggesting that IFN-αfrom pDCs may play a role during the early phase of disease progression. Normally, pDCs have safeguards to avoid the undesirable recognition of self-nucleic acids, including the positioning of TLR inside the cells, rapid decomposition of self-nucleic acid through DNases and RNases, and architectural differences between viral and human nucleic acids [84,85].

mDCs are activated by diverse cytokines, including IL-6, TNF-α, and IFN-α, as well as LL-37–RNA complexes. Activated mDCs secrete a mass of inflammatory cytokines such as IL-12 and IL-23 in psoriasis [86,87], which is indispensable for the proliferation and survival of T cells that produce IL-17. Moreover, Lowes et al. showed that 6-sulfo LacNAc dendritic cells, which specifically accumulate in psoriatic skin lesions, respond to the complex formed between LL-37 and self-RNA via the TLR7 signal, inducing the secretion of IL-17, IL-22, TNF-α, and IFN-γ from Th1/Th17 cells more potently than from other dendritic cells [88].

### 3.4. T Cells and AMPs

Many investigators have demonstrated that psoriasis lesions contain increased numbers of T cells [86]. T cells in psoriasis are activated by IL-23 secreted from activated DCs and macrophages [89]. The Th17 cytokines IL-17A and IL-22 are inducers of hBD2 [90]. Lande et al. demonstrated that approximately two-thirds of patients with psoriasis had CD4^+^ or CD8^+^ T cells which responded to LL-37 [91]. These cells express cutaneous lymphocyte antigen (CLA) and receptors such as CCR6 and CCR10 and secrete IFN-γ and IL-17. The presence of circulating LL-37-specific T cells correlates significantly with disease activity, suggesting a contribution to disease pathogenesis [91]. On the other hand, Peric et al. reported that IL-17A promotes the secretion of cathelicidin from keratinocytes [92], implying that keratinocytes and T cells are critical for positive feedback (Figure 3).

## 4. Cytokine and Intracellular Signaling Regulation by AMPs in Psoriasis

Th17 and Th1 cytokines play roles in chronic inflammation in psoriasis. AMPs enhance the production of some of these cytokines and modulate receptors and cellular signaling in psoriasis.

### 4.1. Interferon and AMPs

Type 1 interferons (IFN-α and IFN-β) are key cytokines that activate autoimmunity, such as systemic lupus erythematosus, and are activated in response to viral infection [93]. IFN-α and IFN-β are suggested to play an indispensable role in initiating psoriasis during skin injury [94]. Type I IFN is important for the pathogenesis of psoriasis and activates autoimmune T cells through the differentiation of dendritic cells [94,95]. Zhang et al. showed that, while IFN-α is primarily produced by pDCs in the dermis, IFN-β is predominantly produced by epidermal keratinocytes in skin wounds and psoriasis lesions [69]. The secretion of IFN-β from keratinocytes promotes the activation and maturation of classic dendritic cells, leading to subsequent T cell proliferation and autoimmune amplification. Furthermore, keratinocyte-derived IFN-β can also promote pDC maturation and activation, suggesting that keratinocytes might also contribute to pDC activation during the early phase of skin injury [69].

Type 1 IFNs can be induced following the activation of endosomal TLR7 and TLR9, or cytosolic cGAS-STING (cyclic GMP-AMP synthase-stimulator of interferon genes), by host, viral, or bacterial DNA. Type 1 IFNs are also induced by the activation of endosomal TLR8 by ssRNA, by the activation of endosomal TLR3, mitochondrial RIG1 (retinoic acid-inducible gene and MAVS (mitochondrial antiviral-signaling protein) by host or viral dsRNA, and by the activation of plasma membrane TLR4 by bacterial lipopolysaccharide (LPS) [96]. Cell responsiveness to various DAMPs relies on the expression of these pattern recognition receptors (PRRs). pDCs express high levels of TLR7 and TLR9, therefore pDCs can rapidly sense self-DNA released upon injury, and then produce IFNα [13,26,39]. TLR4 and TLR8 are usually not expressed in pDCs but are highly expressed in conventional DCs or monocytes [97], making these cells highly responsive to bacterial LPS and self-RNA. In contrast to myeloid-derived immune cells, keratinocytes express high levels of TLR3 and MAVS, but not TLR4, -7, -8, or -9 [42,68,69]. Therefore, keratinocytes rapidly produce IFN-β in response to dsRNA. Zhang et al. showed that wounded keratinocytes upregulate the expression of antimicrobial peptide LL-37, which then enables MAVS and TLR3 in keratinocytes to recognize dsRNA released from dying cells. Keratinocyte-derived IFN-β then promotes DC maturation and subsequent T cell activation to facilitate the development of an autoimmune cutaneous inflammatory response [69]. We demonstrated that LL-37 enables keratinocytes and macrophages to recognize self-non-coding RNA by facilitating binding to cell surface scavenger receptors, enabling recognition by nucleic acid PRRs within cells in human psoriatic skin [98]. These results show that the cell-type-specific expression of pattern recognition receptors shapes the unique and situation-specific innate immune responses of these cells, and that blocking agents or monoclonal antibodies against scavenger receptors may have potential for treating psoriasis.

### 4.2. TLRs and AMPs

Lande et al. showed that LL-37 enhances the recognition of self-DNA through TLR9 in pDCs and enhances inflammation [84]. This overturns the conventional concept of TLR recognizing unmethylated bacterial DNA and inducing an immune response during bacterial infection, indicating that “antimicrobial” peptides are important for the pathogenesis of a non-infectious inflammatory skin disease, psoriasis. TLR9 signal potently induces type I IFN production from pDCs, resulting in activation of mDCs and keratinocytes, and differentiation of Th1/Th17 lymphocytes [9]. β-Defensins and lysosomes were subsequently reported to activate pDCs by enhancing self-DNA or self-RNA recognition, similar to LL-37 [99]. Extracellular RNA complexes are present in psoriatic skin and are associated with mDC activation [97]. LL-37 binds not only to self-DNA but also to self-RNA, and activates TLR7 in pDCs and TLR8 in mDCs [97]. These observations imply a mechanism in which these antibacterial peptides inhibit immunological tolerance to self-antigens, resulting in inflammation in psoriasis.

Morizane et al. reported that LL-37 and extracellular DNA strongly induce type I IFN production from keratinocytes via TLR9 [100]. There are far more keratinocytes than pDCs in the skin, and most self-DNA is likely released from the epidermis because the epidermis is located in the outermost layer of the body where external stimuli, injuries, and inflammation occur most frequently. In light of this, the source of type I IFN in psoriatic skin via the TLR9 signal mediated by cathelicidin and extracellular DNA is questionable.

### 4.3. Scavenger Receptors and AMPs

It remains unclear how extracellular self-RNA released from injured cells reaches intracellular TLR3. We found that the complex formed by LL-37 and dsRNA binds to scavenger receptors and causes uptake of the complex into the intracellular space [98]. We also demonstrated that LL-37 and similar α-helical AMPs can form pro-inflammatory nanocrystalline complexes with dsRNA that are recognized by TLR3 differently from dsRNA alone [101]. Self-nucleic acids (DNA, RNA) are DAMPs released from damaged cells and extracellular matrix to activate the innate immune system. Psoriasis shows the Köbner phenomenon. Inflammation induced by DAMPs released due to microinjuries or scratches, and the enhanced recognition of DAMPs by antimicrobial peptides, may be involved in the mechanism underlying the Köbner phenomenon [94].

## 5. Involvement of AMPs in Clinical Aspects of Psoriasis

Current genetic analysis tools and methodologies are lending insights into the genetic background of psoriasis, and molecular biology techniques are providing effective biologics for psoriasis treatments. These technological developments have shown promising links between AMPs in clinical phonotypes and psoriasis treatments.

### 5.1. Psoriasis Phenotypes and AMPs

Psoriasis is thought to be triggered by both genetic and environmental factors. Strong association between psoriasis and the human leukocyte antigen (HLA)-C*06:02 allele has been reported in various races. Mabuchi et al. predicted sequences of peptides which are included in LL-37 and binds strongly to HLA-*C06:02 using computer models [102]. Interestingly, some peptides were shown experimentally to combine with HLA-*C06:02 [91]. Furthermore, Mabuchi et al. proposed that the peptides bind strongly to HLA-*C06:02 but not to T cell receptor (TCR). Such peptides may provide new drugs for the treatment of psoriasis by inhibiting the binding between HLA-*C06:02 and TCR.

Yuan et al. found that the levels of anti-LL-37 and anti-ADAMTS-L5 autoantibodies were significantly elevated in patients with PsA compared to non-PsA controls, suggesting that these molecules may be involved in the pathogenesis of PsA [103]. Frasca et al. found that LL-37 and autoantibodies to LL-37 are elevated in the synovial fluid (SF) of patients with PsA but not in osteoarthritis patients. Anti-carbamylated/citrullinated-LL-37 antibodies are present in the SF and plasma of PsA patients, and at lower levels in the plasma of psoriasis patients, but not in controls. Furthermore, plasma anti-carbamylated-LL-37 antibodies correlate with PsA (DAS44) but not psoriasis (PASI) disease activity. Frasca et al. proposed that LL-37 is a novel PsA autoantibody target, and that plasma antibodies to carbamylated-LL-37 are new PsA activity markers [47].

Narrowband-UVB treatment induces vitamin D production and subsequent secretion of cathelicidin [104]. Cyclosporine [105], etanercept, and anti-TNF-α antibody [106] reduce or suppress the expression of cathelicidin. LL-37 inhibits the apoptosis of keratinocytes and dermal capillary endothelial cells and stimulates their proliferation [107].

### 5.2. Psoriasis Treatments and AMPs

A particularly surprising observation came with the recognition that the human cathelicidin gene is under transcriptional control of a vitamin D response element (VDRE) [108,109]. Following skin injury or infection, 25(OH)D3 is hydroxylated by the enzyme cytochrome p450 27B1 (CYP27B1) to 1,25(OH)_2_D3. The reaction is stimulated locally by the activation of TLR2 or local cytokines, such as TNF or type I IFNs [110,111]. This local enzymatic event enables the rapid induction of CAMP expression through the binding of 1,25(OH)_2_D3 to VDRE. These observations suggest that AMP expression might be influenced by serum vitamin D levels [112], dietary vitamin D [113], or vitamin D generated by skin exposure to sunlight [114]. This means that nutritional intake probably provides important signals that control AMP expression. Conversely, LL-37 transactivates epidermal growth factor receptor and downstream signaling in epithelial cells [115,116].

Vitamin D is important in the relationship between psoriasis and cathelicidin. Vitamin D derivatives are used externally to treat psoriasis and may promote cell differentiation and suppress proliferation. Recently, vitamin D was reported to suppress the expression of hBD2, hBD3, IL-17A/F and IL-8 in psoriasis plaques [117] and induce CD4^+^CD25^+^ regulatory T cells [118]. These results suggest that vitamin D affects immunological aspects of psoriasis. Vitamin D3 strongly induces cathelicidin secretion from keratinocytes and monocytes [118], which is inconsistent with LL-37 generally acting to induce inflammation. However, Dombrowski et al. reported that intracellular LL-37 inhibits the formation and activation of DNA sensor AIM2 inflammasomes and subsequent secretion of IL-1β, resulting in an anti-inflammatory effect [119]. Whether LL-37 and DNA induce or suppress inflammation may depend on their localization (inside the cytoplasm or taken up from the extracellular space) [100].

Anti-TNF agents are highly effective in the treatment of psoriasis, but 2–5% of treated patients develop psoriasis-like skin lesions called “paradoxical psoriasis” [120]. Conrad et al. showed that skin lesions from patients with paradoxical psoriasis are characterized by a selective overexpression of type I IFN, dermal accumulation of pDC, and reduced T cell numbers, when compared to patients with classic psoriasis [121].

## 6. Conclusions

Psoriasis is not just an inflammation of the epidermis or an immune disease derived only from T cells, but a systemic inflammatory disease caused by crosstalk between various cells such as keratinocytes, neutrophils, dendritic cells, and T cells. Contrary to their name, “antimicrobial” peptides are often involved in the pathogenesis of psoriasis by activating the innate immune system and triggering inflammation by various mechanisms independent of infection. At present, there is no established treatment targeting antimicrobial peptides, and blocking or degradation of AMPs is expected to be a novel treatment for psoriasis.

## Figures and Tables

**Figure 1 ijms-21-06791-f001:**
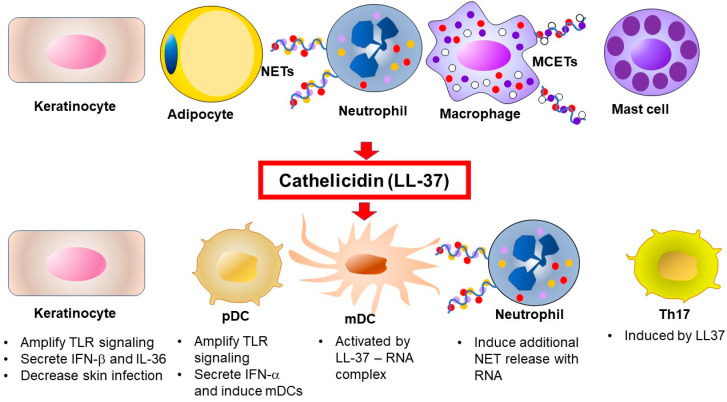
Source and function of cathelicidin in psoriasis. Once inflamed, skin produces cathelicidins through increased expression by keratinocytes and adipocytes, and its local deposition is increased by recruited neutrophils. The active form of cathelicidin, LL-37, activates immune cells via multiple mechanisms in psoriasis. TLR, Toll-like receptor, NET, neutrophil extracellular trap, MCET, mast cell extracellular trap, pDC, plasmacytoid dendritic cell, mDC, myeloid (conventional) dendritic cell, Th, T-helper cell, IL, interleukin, IFN, interferon.

**Figure 2 ijms-21-06791-f002:**
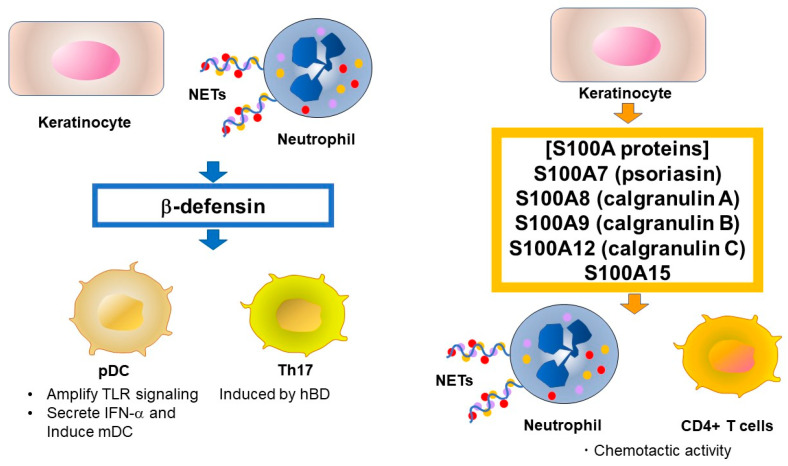
Source and function of human β-defensin and S100 proteins in psoriasis. β-Defensins and lysosomes were reported to activate pDCs by enhancing self-DNA or self-RNA recognition, similar to LL-37. S100A7 has chemotactic activity for neutrophils and CD4^+^ T lymphocytes. TLR, Toll-like receptor, NET, neutrophil extracellular trap, pDC, plasmacytoid dendritic cell, Th, T-helper cell, IFN, interferon.

**Figure 3 ijms-21-06791-f003:**
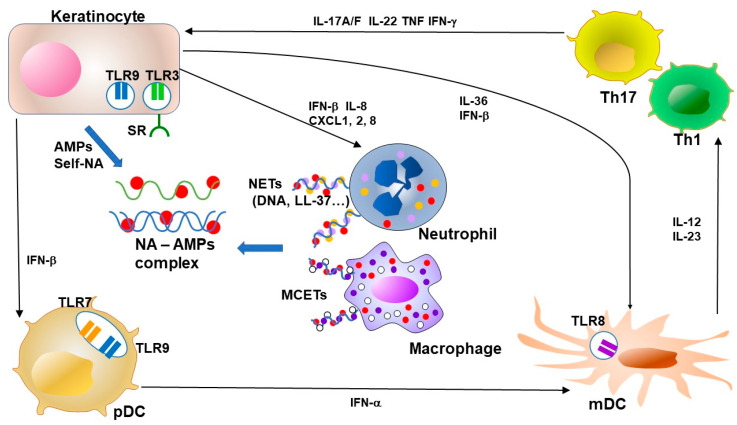
Feedback loops of AMPs and NA–AMP complexes in psoriasis. Keratinocytes in psoriasis show overproduction of antimicrobial peptides (AMPs). NETs from neutrophils and MCETs from macrophages form complexes with NA and AMPs. AMPs enhance the recognition of viral or autologous nucleic acids (DNA and RNA) through TLR in keratinocytes and dendritic cells. The TLR signal potently induces type I IFN production from keratinocytes and plasmacytoid DC (pDC). LL-37 enables keratinocytes and macrophages to recognize RNA by facilitating binding to cell surface scavenger receptors that enable recognition by TLR3. Keratinocyte-derived IFN-β promotes pDC and mDC maturation. pDC-derived IFN-α induces activation and maturation of mDC. LL-37 further stimulates keratinocytes to produce IL-36s and other cytokines as an alarmin function. Activated mDCs secrete a mass of inflammatory cytokines, including IL-12 and IL-23 in psoriasis, and this secretion is indispensable for the expansion and survival of T cells that produce IL-17. Cytokines from Th17 and Th1, including IL-17, IL-22, TNF and IFN-γ, accelerate the proliferation of keratinocytes and the secretion of chemokines and AMPs from keratinocytes. NA, nucleic acid, AMP, antimicrobial peptide, TLR, Toll-like receptor, NET, neutrophil extracellular trap, MCET, mast cell extracellular trap, pDC, plasmacytoid dendritic cell, mDC, myeloid (conventional) dendritic cell, Th, T-helper cell, IL, interleukin, IFN, interferon, TNF, tumor-necrosis factor.

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
