# Peer review of "Psoriasis and Antimicrobial Peptides"

_ijms, 2020, doi:10.3390/ijms21186791_

Round 1
Reviewer 1 Report
Well prepared review, covering the topic. Actual, interesting.
Comments:
- Abstract is short, should be more specified
- The whole Chapter 3 is very detailed, describing plenty of data which very often are not close to psoriasis. My recommendation is to make this part shorter
- Chapter 4 is comprehensive, focused on the topic and well done
- Very important part is the description of relation between psoriasis and vitamin D metabolism, I appreciate it
Author Response
Well prepared review, covering the topic. Actual, interesting.
I appreciate your kind comments.
Comments:
Abstract is short, should be more specified
I added sentences in the abstract as the reviewer suggested to improve the Abstract section.
The whole Chapter 3 is very detailed, describing plenty of data which very often are not close to psoriasis. My recommendation is to make this part shorter
I appreciate your suggestion. I agree that Chapter 3 is a little lengthy, especially for expert dermatologists or immunologists. But I think this part is important for many of readers to recognize that antimicrobial peptides have not only antimicrobial function but also immunological function. Therefore, I’d like to keep this chapter as it is. Thank you again for your valuable advice.
Chapter 4 is comprehensive, focused on the topic and well done
Very important part is the description of relation between psoriasis and vitamin D metabolism, I appreciate it.
Thank you for your kind comments.
Reviewer 2 Report
Toshiya Takahashi and Kenshi Yamasaki presented a comprehensive summary of antimicrobial peptides in psoriasis. The study is intriguing, and the references are solid. I have a few minor questions as follows:
- The functions of each peptide category is well described. In my opinion, a figure to summarize and compare the different roles of peptides in psoriasis is missing.
- Beside the ones presented here, are there any other promising peptides under investigation? If the information is added, is will be interesting.
- How about the pros and cons of different categories?
Author Response
Toshiya Takahashi and Kenshi Yamasaki presented a comprehensive summary of antimicrobial peptides in psoriasis. The study is intriguing, and the references are solid. I have a few minor questions as follows:
The functions of each peptide category is well described.
I thank you for your kind comments.
In my opinion, a figure to summarize and compare the different roles of peptides in psoriasis is missing.
I appreciate your sharp pointing out. I agree that the information is missing to compare the different roles of each peptide in psoriasis. Since no article compared functions of antimicrobial peptides side by side regarding to the psoriasis pathogenesis, it is very hard to summarize and compare the roles of peptides in psoriasis pathogenesis. Actually, I have once tried to make a such kind of a figure to compare functions of each antimicrobial peptides before the original submission. But we felt the figure will be unfair to present for readers because the volumes of the source information are not equally provided from each AMP. Therefore, we appreciate the reviewer for understanding that we do not provide the figure as suggested. Thank you again for your kind advice.
Beside the ones presented here, are there any other promising peptides under investigation? If the information is added, is will be interesting.
Thank you for your shrewd comment. I added description of “other AMPs” (page 5, line 163-168).
How about the pros and cons of different categories?
This is a very interesting question. As far as we understand from reports of articles, pros of AMPs in psoriasis is antimicrobial function to protect against infectious pathogens in inflamed skin. Cons are the proinflammatory functions of AMPs to exacerbate psoriatic skin inflammation. We do not know articles with enough information to tell differences of pros and cons of different categories. Thank you for your very interesting suggestion.
Reviewer 3 Report
The review is well-written and clear, and gives a good overview of the available literature. The cited works include important recent contributions, as well as some of the important older work that formed the basis of newer results.
The first two figures are easily understandable and give a good overview of the role of various AMPs in psoriasis. The third figure is a bit too crowded and it is less clear what message the authors were trying to send, I would recommend simplifying it.
The only fault I could find is that the formatting of the text is not universal, since some parts are single spaced while most are double, but I assume this will be fixed by the editors.
I would recommend briefly mentioning other AMPs cleaved from hCAP18, or at least listing them, even if these do not have any proven role in psoriasi, to make the manuscript more complete.
Author Response
The review is well-written and clear, and gives a good overview of the available literature. The cited works include important recent contributions, as well as some of the important older work that formed the basis of newer results.
I thank you for your kind comments.
The first two figures are easily understandable and give a good overview of the role of various AMPs in psoriasis. The third figure is a bit too crowded and it is less clear what message the authors were trying to send, I would recommend simplifying it.
I appreciate your sharp pointing out. I rearranged components in Figure 3 (Figure 3).
The only fault I could find is that the formatting of the text is not universal, since some parts are single spaced while most are double, but I assume this will be fixed by the editors.
Thank you for your suggestion. I emended line space in the manuscript.
I would recommend briefly mentioning other AMPs cleaved from hCAP18, or at least listing them, even if these do not have any proven role in psoriasis, to make the manuscript more complete.
I thank you for your considerate comment. I added description of hCAP18-derived peptides (page 3, line 103-106).
Reviewer 4 Report
Psoriasis is not only a skin and joint disease but a systemic inflammatory disorder caused by crosstalk between various cells. The manuscript is an exhaustive review paper considering psoriasis and antimicrobial peptides (AMP).
Antimicrobial peptides (AMP) have been identified in almost all plants and animals. Recent studies revealed their role in the human skin and skin pathologies. The article summarizes the role of AMPs in the psoriatic lesions. The authors emphasize not only the „antimicrobial” role of the AMPs but also their role in the immunity: production of some cytokines, modulation of receptors, cellular signaling in psoriasis.
This is well written, very interesting review article presenting the current level of knowledge.
Author Response
Psoriasis is not only a skin and joint disease, but a systemic inflammatory disorder caused by crosstalk between various cells. The manuscript is an exhaustive review paper considering psoriasis and antimicrobial peptides (AMP).
Antimicrobial peptides (AMP) have been identified in almost all plants and animals. Recent studies revealed their role in the human skin and skin pathologies. The article summarizes the role of AMPs in the psoriatic lesions. The authors emphasize not only the „antimicrobial” role of the AMPs but also their role in the immunity: production of some cytokines, modulation of receptors, cellular signaling in psoriasis.
This is well written, very interesting review article presenting the current level of knowledge.
I appreciate your kind comments.